# Community Training Institute for Health Disparities: Outcomes of a Formal Opportunity for Community Capacity Building to Increase Health Equity in Southern Puerto Rico

**DOI:** 10.3390/ijerph22010080

**Published:** 2025-01-09

**Authors:** Julio Jiménez-Chávez, Fernando J. Rosario-Maldonado, David A. Vélez-Maldonado, Dorimar Rodríguez-Torruella, Jeannie M. Aguirre-Hernández, Eida Castro-Figueroa, Gloria Asencio-Toro, Elizabeth Rivera-Mateo, Luisa Morales-Torres, Axel Ramos-Lucca, Jorge L. Motta-Pagán, Nina Wallerstein, Melissa Marzán-Rodríguez

**Affiliations:** 1School of Behavioral and Brain Sciences, Ponce Health Sciences University, Ponce, PR 00716, USA; ecastro@psm.edu (E.C.-F.); gasenciotoro@gmail.com (G.A.-T.); axelramos@psm.edu (A.R.-L.); 2Ponce Research Institute, Ponce Health Sciences University, Ponce, PR 00716, USA; frosario@psm.edu (F.J.R.-M.); davelez@psm.edu (D.A.V.-M.); dorodriguez@psm.edu (D.R.-T.); jaguirre20@stu.psm.edu (J.M.A.-H.); erivera@psm.edu (E.R.-M.); lmorales@psm.edu (L.M.-T.); jomotta@psm.edu (J.L.M.-P.); mmarzan@psm.edu (M.M.-R.); 3Public Health Program, Ponce Health Sciences University, Ponce, PR 00716, USA; 4College of Population Health, University of New Mexico, Albuquerque, NM 87131, USA; nwallerstein@salud.unm.edu

**Keywords:** community capacity building, health disparities research, health promotion, community-based participatory research, community researchers, community health promoters, Hispanic

## Abstract

Integration of the community into health research through community-engaged research has proven to be an essential strategy for reducing health inequities. It brings significant benefits by addressing community health concerns and promoting active community participation in research. The Community Training Institute for Health Disparities (CTIHD) was established to support this integration based on Community-Based Participatory Research (CBPR) principles. The main objective of this paper is to report the CTIHD program’s implementation, evaluation, and outcomes from the first two cohorts. The CTIHD recruited Hispanic community members (*N* = 54) to be trained in health disparities research and health promotion to foster Community–Academic Partnerships (CAPs) and develop community-led health promotion interventions. Evaluation measures included satisfaction, knowledge change, retention rate, completion rate, and project proposals (research and community health promotion plans). The retention and completion rates were 83% and 78%, respectively, with forty-two (*n* = 42) community trainees receiving the completion certification. Both cohorts demonstrated a significant increase in knowledge (*p* < 0.05), and overall satisfaction exceeded 90%. Outcomes include seven (7) community–academic partnerships, leading to the co-development of research proposals, three (3) of which received funding. Additionally, twenty-two (22) community health promotion plans were developed, with seven (7) implemented, impacting 224 individuals. Findings from this study suggest that the CTIHD effectively provided capacity building, promoted the formation of CAPs, and increased community-led health promotion interventions, thereby advancing health disparity research and community initiatives.

## 1. Introduction

For more than two decades, Community-Engaged Research (CEnR) has proven to be a helpful strategy in the health field, especially in research focused on reducing health inequities and improving community health outcomes [1]. A proliferation of literature has emerged during this period on Community-Based Participatory Research (CBPR) as critical research focused on developing the CEnR continuum [2,3,4]. Broad evidence has been documented about the benefits of CBPR in improving the response to real community concerns about their health problems and the applicability of research findings to their needs, increased confidence in the research field and researchers, and reduced health inequities [5].

Healthcare system stakeholders, funding agencies, and community-driven initiatives with evidence-based health interventions have increasingly implemented Community Engagement (CE) approaches [6]. An example of this is the extensive reports of results from implementing studies based on CBPR to address health problems related to the prevention and control of chronic health conditions, such as diabetes, cardiovascular diseases, respiratory diseases, mental health, and infections [7,8,9,10,11,12]. Promoting community–academic partnerships (CAPs) has been one of the main focuses of CBPR [13,14]. These associations require preparation, a high level of commitment, cooperation, and the ability to negotiate for the benefit of all parties, always prioritizing the collective’s health needs [15]. In a CAP, experiential and academic knowledge represents a valuable instrument to face challenges that may arise throughout the research process, including implementation and dissemination [6].

An essential characteristic of CBPR, as a research approach aimed at reducing disparities, is its ability to recognize the impact and effects of social determinants on health outcomes [16,17]. Health outcomes cannot be separated from the relationship between social, political, and economic factors that impact behaviors and resources that influence health [18]. CBPR promoters describe it as a research approach based on an equitable partnership between community and academia where experiences, responsibilities, and decision making are shared to address a specific problem that impacts the health of groups of individuals [15,19].

Yet equitable partnerships are usually challenging to realize in practice. Power imbalances exist when academic knowledge is often perceived to be more worthwhile than community knowledge or when community members frequently face more significant challenges in devoting time to research projects, with academic members’ time more covered by grant funding. Increasingly, there has been a call to promote community power and voice in the research enterprise to arrive at equitable power in research decision making and the community’s use of research results to generate social change, social justice, and community benefits [20,21,22,23]. Community participation can promote equity in health, based on the idea that not only a part of an individual’s health status depends on their behavior or habits. Other factors to consider are poverty, unemployment, low education level, poor public transportation, geographic location, and inadequate housing conditions as social determinants associated with health outcomes [24,25,26]. The community’s participation often can best elucidate these factors and how they must be incorporated into the research enterprise.

### 1.1. Theoretical Frameworks

Conceptual models that seek to explain the bases of health inequities integrate community solutions to promote equity in health. Common principles include as follows:Making health equity an issue that represents a shared social value.Increasing community capacity and participation in promoting positive health outcomes.Increasing multisectoral/multilevel collaborations as critical strategies to reduce health inequalities and improve community health outcomes.

The model proposed by the National Academies of Sciences, Engineering, and Medicine, Division of Health, and Medicine, emphasizes inequities as barriers to promoting community health [27]. The Committee on Community-Based Solutions to Promote Health Equity established the available review literature to examine the status, causes, and conditions related to health disparities in the United States of America that disproportionately affect underserved populations. The proposed model adapts elements of the Robert Wood Johnson Foundation’s Culture of Health Action Framework [28] and the Prevention Institute’s Systems Framework to achieve an Equitable Culture of Health [29,30]. This circular model recognizes the complexity of the problem and accepts the reality of structural inequities in health systems, social determinants, and community-led solutions [27].

In 2013, the revised version of the CBPR Conceptual Model was updated, integrating the community perspective collected from a series of focus groups, confirming its community validity [31]. This version of the model reaffirms that the relational partnership (i.e., trust and dialogue) and structural processes (i.e., formal agreements and sharing of resources) produce a critical equitable dimension cited between context (social determinants, politics, culture) and research/interventions (research questions and design, co-learning experience) [31,32]. Indeed, these models postulate that when communities affected by a health problem are integrated into research, the possibility of developing and implementing successful interventions and the potential for diffusion within the community increases [33,34]. The benefits of improving community health outcomes and the strategies developed for this purpose must be maintained over time. A trained community can lead to the sustainability of these results [35,36]. However, community members deserve to be capacitated to participate in selecting the research topic with their priorities, collaborate with academics in developing research questions and objectives, co-interpret the findings, and co-disseminate/implement research results [37]. Community (and academic) capacity building is a fundamental axis in CPBR and can be a valuable strategy to achieve equitable partnering and sustainability of the results obtained during the investigation. Capacity building and sustainability are interconnected concepts [38].

Undoubtedly, the voices and perspectives of the communities must be integrated into all phases of the research process to incorporate their cultural characteristics and knowledge of their values, share their health concerns, and thus achieve beneficial results that respond to their actual needs [38,39]. However, it can be paradoxical to expect or demand active community participation in research without providing formal training spaces for the lay community. Creating training opportunities for members of underserved communities is also a matter of social justice.

The Community Engagement Core of the Center for Health Disparities at the Ponce Health Sciences University (PHSU) focuses on developing strategies to reduce health disparities related to chronic diseases in low-income populations. Backed by extensive evidence of the benefits of integrating the CBPR approach to identify needs and develop intervention models to improve health outcomes, the Community Training Institute for Health Disparities (CTIHD) was created. The CTIHD’s catchment area of influence is in southern Puerto Rico (PR), where the predominant language is Spanish. With the creation of the CTIHD, we planned to establish an innovative and sustainable structure in PR, co-developed with members of the community and supported by a health-focused university. Our purpose was to examine whether the development and implementation of a formal training institute tailored to community members would be an effective strategy for acquiring knowledge and skills that facilitate the formation of partnerships with academic researchers and disseminating health education to peers.

This manuscript’s main objective is to report on the implementation, evaluation, and results of the CTIHD program’s first two cohorts.

### 1.2. Background

#### Community Training Institute for Health Disparities: Goals and Design

At the PHSU Specialized Center in Health Disparities, the CTIHD’s primary goal is to increase health disparity research and improve community health outcomes through strengthening community and academic partnerships and health education to achieve health equity, according to identified community health needs in low-income Hispanic populations. This goal is accomplished by establishing two programs to train the community: (a) the Community Researcher Program (CRP), aimed at capacitating with basic research knowledge and skills to increase health disparities research, and (b) the Community Health Promotion Program (CHPP) aimed at training health promoters to increase community health promotion. Over the 5-year cycle, the CTIHD has an outcomes-based evaluation aligned with the project’s logic model to measure the results of the capacitation programs. Considering the CTIHD’s primary goal, the program’s curriculum designs and learning model were identified as problem-centered and competency-based [40].

The program’s primary outcomes consisted of participants acquiring the basic skills and knowledge required for co-developing research proposals with academic researchers or creating community health promotion plans focused on reducing health disparities, depending on the program enrolled. The programs were carried out from October 2020 to February 2024.

The PSHU-RCMI-CEC team comprises six (6) academic researchers and thirteen (13) trained community partners (Community Trained Workforce [CTW] and Community Scientific Advisory Committee [CSAC]. The CSAC reviewed and approved the final version of the curriculum and syllabuses. After review by the IRB Committee, following Federal regulations, 45 CFR Part 46.101(b) (7), the presented study protocol was approved and considered exempt due to the research being conducted in a commonly acceptable setting (e.g., university or online classroom setting), and standard educational practices were utilized (Protocol #: 2201083931). Other articles discuss these programs’ conceptualization, development, and implementation [41].

## 2. Methods: Program Implementation and Evaluation

### 2.1. Conceptual Model and Program Design of CTIHD Programs

The CTIHD educational programs are guided by a problem-centered curricular design and competency-based learning model grounded on CBPR principles. Problem-centered teaching incorporates real-world problems to drive trainees to learn concepts and principles. This promotes critical thinking, problem solving, and communication skills while providing opportunities for working in groups, finding and evaluating research materials, and life-long learning. Additionally, utilizing a competency-based curriculum emphasizes what learners are expected to do rather than mainly focusing on what they are expected to know [40].

#### Capacity Building Coursework

Community trainees from each cohort of the CRP and CHPP received six (6) courses and two (2) courses, respectively, over eight (8) months. Each 10-session course was conducted remotely (due to the COVID-19 pandemic) and was facilitated by knowledgeable instructors with experience in CE.

In both programs, the courses consisted of didactic sessions (2 h) and practical activities (1 h). To ensure consistency across courses, instructors were provided with rubrics that included assigned topics, course objectives (length, format, and content), hands-on activities, and a question guide to assess knowledge acquired (including the number of questions and problems to be considered when writing those questions).

### 2.2. Community Trainees Recruitment and Sampling

The two cohorts of the CTIHD were comprised of a purposeful sampling of *N* = 54 community trainees (of the 111 who applied). Additionally, as established by the Admissions Committee (comprised of the research team and CTW), each program selected 12–15 community trainees per cohort. Applicants had to meet the following requisites: (1) 21 years or older, (2) a minimum of a high school degree, (3) self-identified as Hispanic, and (4) be a community member in a municipality of the southern region of Puerto Rico (catchment area of the CTIHD). Individuals who applied and were interviewed for the program were selected by the Admission Committee, following established selection criteria which were goals or objectives for participating in one of the CTIHD programs, experiences working with communities, teamwork experience, availability of time, and letters of recommendation. The Admissions Committee explored these criteria through the interview process and the documentation review provided in the application process; these were evaluated and scored utilizing developed evaluation criteria sheets. Completion and certification required community trainees to attend no less than 70% of training sessions, complete the practical assignments, and co-conceptualize and co-develop a final project (CBPR research proposal with partnered academic researchers or a community health promotional plan, depending on the program enrolled). All educational training and support materials were free of cost. For the overall process, see Figure 1.

The program was promoted through community partners, collaborators, institutional social media, flyers, community health fairs, and a Virtual Open House activity (the requirements and selection criteria were included in all promotions). The promotional activities ensured a wide area of capture and equal opportunities for interested applicants in under-resourced communities. Potential trainees were given respective orientations of both CTIHD programs, including overall program descriptions, program structure and schedules, and completion requirements. Orientations were provided through the phone, community activities, video conference platforms, and in-person peer potential candidates who showed interest.

#### 2.2.1. Evaluation and Data Analysis: Knowledge Change, Satisfaction, Retention Rate, Completion Rate, and Project Proposals

A mixed-methods design was developed to evaluate the CTIHD utilizing measurement tools such as pre- and post-tests, course evaluation, cognitive debriefing (at the end of the program), and program documentation. Knowledge change was evaluated through pre- and post-tests, administered at each course’s beginning and end. Each instructor developed multiple choice questions using, as a reference, the overarching information presented in the course; the questions were presented to and approved by the research team to verify their comprehensibility (i.e., ensure no vague or ambiguous questions) and fidelity of the rubric. Evaluation forms were administered post-course to receive participants’ feedback for optimizing courses while measuring satisfaction and perceived readiness to apply knowledge. The participants completed a cognitive debriefing session to assess the curriculum’s acceptability, perceived readiness, and utility at the program’s end. The session included questions exploring general thoughts of the program, research proposal or health plan development, perceived readiness to apply knowledge, utility of the program, and general satisfaction.

IBM SPSS Statistics (version 24) was utilized to generate descriptive statistics (e.g., frequency, percentages, averages) of participant characteristics and course evaluations. To assess knowledge change, the non-parametric Wilcoxon Signed Rank Test was utilized. The pretest and post-test were assigned to groups 1 and 2 for the analysis. The normal distribution of the results was assessed using the Kolmogorov–Smirnov test, which confirmed that the obtained data showed a non-normal distribution (*p* < 0.05). Thematic analysis was utilized to analyze qualitative data using transcribed cognitive debriefing discussions [42,43]. Two research assistants reviewed, coded, and identified relevant themes. After initial review and analysis, codes and themes were discussed with the principal investigator to review and resolve any conflicting codes or themes.

Retention Rate (%):(retained participants during the program implementation/n) × 100

Completion Rate (%):(community trainees who met completion requirements of program/n) × 100

#### 2.2.2. Research Proposal or Community Health Promotional Plan Development

During the final trimester of each program, community trainees from both programs were provided with an orientation of the projects to be co-developed to complete their respective programs and integrate the knowledge gained in the courses. Each program orientation included (1) an explanation of documentation to guide students in the project’s development, (2) an explanation of rubrics for evaluating the final product, and (3) an introduction of mentors to aid trainees in the final project development. Additionally, (4) they were provided with a timeline to complete sections of the projects and meet with their mentors to review their work and provide guidance and support as needed. Mentors had the expertise to provide feedback on project development and partnership guidance during virtual meetings (Zoom Platform).

For the CRP proposal development, community trainees formed collaboration groups based on topics of common community health concerns. These groups were matched with an Academic Researcher (AR) from PHSU who shared their research interest through Community Forums, where community trainees presented their initial ideas to academic researchers.

#### 2.2.3. Community–Academic Forum

This community forum was strategically designed to create a space to link academic researchers with the community health needs and priorities presented by the CTIHD community trainees. The community trainees from the CRP, in groups, presented their research questions and objectives related to their prioritized health issues in their communities to academic researchers. We consider this a pathway for bringing community voices to academia and connecting community needs with field interests and academic expertise. For the CHPP, it provided a platform for community trainees to identify areas for health promotion.

After the matching in the community forum, during the proposal development, community trainees from the CRP shared experiential knowledge (e.g., the viability of the project, culture in their communities, and best strategies to collect data), and ARs shared scientific knowledge (e.g., objective creation and research design) to co-develop a community-centered research proposal.

For the CHPP, community trainees worked individually on developing their community health promotion action plans. This individualized approach allowed each trainee to tailor their plans to address the unique needs of their respective communities. The work was guided by public health practitioners from the RCMI-CEC with extensive experience in health education, who provided critical support, including expert instruction, mentorship, and regular feedback to ensure the plans were practical and impactful. The structure also aligned with the community trainees’ roles as community health promoters, where they are expected to conduct their work autonomously and effectively address localized health issues. The individualized format ensured community trainees could apply their problem-solving skills in a real-world context while receiving tailored academic support. Trainees shared valuable feedback and insights with the practitioners, contributing to the iterative improvement of the program and fostering personal and professional growth.

## 3. Results

### 3.1. Sociodemographic Data, Acceptance Rate, and Retention Rate

Table 1 demonstrates sociodemographic results. Fifty-four (n = 54) participants were recruited from applicants into the CTIHD programs with an overall acceptance rate of 49%. Nine dropouts were evidenced during the implementation phase, resulting in an 83% overall retention rate. Participants reported being part of various groups or sectors such as international health organizations (e.g., Doctors Without Borders), professional local groups (e.g., nursing professional colleges), local community- and faith-based organizations, and local government. Figure 2 demonstrates the catchment area of the municipalities where trainees were recruited.

### 3.2. Knowledge Change

Overall, both programs demonstrated a statistically significant increase in knowledge gained by participants in both individual cohorts and pooled results. Additionally, cohort 2 of both programs displayed increased median percentages in the post-test compared to cohort 1, see Table 2.

### 3.3. Satisfaction: Course Evaluations

Satisfaction was collected quantitatively and qualitatively. Through evaluation forms (quantitative), the overall satisfaction obtained was greater than 90% for all cohorts and programs (see Table 3).

### 3.4. Cognitive Debriefing

At the end of the program, community trainees from each cohort were invited to evaluate their programs once they had completed all requirements. The evaluated criteria organized the themes and sub-themes identified from the cognitive debriefing sessions. (see Table 4).

### 3.5. Developed Pre-Pilot Proposals and Community Health Promotional Plans, and Completion Rate

Forty-two (42) community trainees completed their expected pre-pilot proposals and community health promotional plans, achieving a completion rate of 78%. Three (3) participants could not complete their final project due to personal reasons. See Table 5 for the expected outcomes of the CTIHD programs.

### 3.6. Expected Outcomes and Implementation of CTIHD Pre-Pilot Proposals and Community Health Promotional Plans

Through the petition and initiatives of the community–academic partnership members, funding mechanisms were explored and identified to implement the pre-pilot proposals. Internal and external sponsors financed three (3) research proposals, which are being implemented. Similarly, community health promoters explored and identified mechanisms for funding the community health promotional plans, with one financed by an external sponsor. Seven community health promotional plans were coordinated and implemented by the trained community, with support from the CTIHD, impacting 224 attendees. Implemented community health promotional plans obtained a 97% satisfaction rate and a statistically significant change in attendees’ knowledge (pre- and post-test; *p* < 0.05)**.**

## 4. Discussion

In previous studies conducted by our research team [37], we collected the interest and willingness of community members to participate in research and health promotion activities to respond to the health concerns of their communities. However, many often recognize and express barriers (e.g., technical language) that limit establishing effective, direct, and fluid communication with academic researchers [38]. Expressions such as “they are up there in the universities, and we are down there, we do not understand them, and they do not understand us” conveyed the need to create a mechanism to bring both perspectives closer together and better understand each side. In 2019, the Community Training Institute for Health Disparities (CTIHD) was created in response to this need.

During these five years, CTIHD developed and implemented two programs to increase research on health disparities and improve community health outcomes through the Community Research Program and the Community Health Promotion Program to achieve health equity in underserved Hispanic populations. CTIHD promotes the formal training of community members to participate and actively integrate in the co-development and co-implementation (community and academic researchers) of projects aimed at addressing the health needs and outcomes of the community. The training offered by the CTIHD allowed community members, many of them with a high school level of academic preparation, to acquire the basic skills and knowledge necessary to develop research proposals in conjunction with academic researchers and create community health promotion plans focused on reducing health disparities. CTIHD completed two cohorts and demonstrated an increase in knowledge change (*p* < 0.05) in both programs, in the case of the second cohort, with a comparatively higher trend than the first. Learners’ satisfaction evaluations with courses and programs generally ranged from 88% to 100%.

While the CBPR approach emphasizes developing equitable partnerships between communities and academics, few programs for the Hispanic population are developed within a formal academic structure. Equally, few are conceptualized within a competency-based model adapted to the general community’s education level and offer certification from a health sciences university such as PHSU. On the other hand, this training model integrates practical experiences that allow moving from the didactic to the experiential within the learning process. The competency-based learning model was appropriate for the program design since it considers an individual’s skills, knowledge, and abilities to train successfully. This model is based on the idea that learning is an active process achieved through practice and experience [40]. Acquiring knowledge and skills, establishing alliances with academics, co-developing proposals with researchers, and implementing research studies that respond to the needs of their communities make up a unique, comprehensive learning experience within the CBPR framework. Moreover, the iterative process of refining the contents of both programs included cognitive briefings. The trainees had the opportunity to express their opinions on the programs’ structure and content. All the trainees’ suggestions were addressed and discussed, and most of them were incorporated into the courses of the CTIHD programs.

During the briefings, all community trainees agreed that the training met or exceeded their expectations, motivating them to continue their community work. They also emphasized that the content of the program’s courses was understandable and that the overall structure and sequence of the curriculum facilitated the learning process. However, some trainees agreed that the Research Methodology course needed further adjustments, such as extending its duration, due to its challenging materials. Participants also noted that developing the CRP research proposal was challenging and enriching for all parties. In response to this feedback, we implemented several strategies: we provided additional mentorship (especially during the early stages of research proposal development), encouraged early identification of research topics, and integrated academic researchers into the CBPR course to promote earlier engagement between researchers and community trainees. When establishing healthy CAPs under the CBPR, it is crucial to train both parties, considering that the CBPR breaks with traditional research paradigms by an equitative distribution of power dynamics [44,45].

For community trainees who expressed training-related stress, an academic counselor (clinical psychologist) was available to support them in managing academic anxiety. For personal situations that negatively impacted their academic performance, referrals were provided to a psychological help center affiliated with the university. Even though stress was expressed, one CRP trainee mentioned: “Simple: I think we already have the tools; we know how to use them. Now, we must use them”. In addition, CRP participants remarked they felt ready to use the tools they acquired in practice to develop the research proposal. In the case of the CHPP participants, the process of creating the community health promotion plan was defined as enriching. However, the trainees indicated the need for more time to discuss the development process and structure of the educational plan. Specific sessions are now provided to develop the educational plan in response to the community trainees’ recommendations. One CHPP trainee stated: “I understand that the program has prepared me a lot. I have spent several years coordinating activities in some communities. This program has helped me strengthen my knowledge to work with the communities …”

The outcomes of the CTIHD include the development of seven (7) research proposals and twenty-two (22) community health promotion plans on various topics focused on community health needs identified by participants. Hitherto, through the CRP, seven (7) partnerships have been formed between the community and academia to develop research proposals; three (3) proposals have received funding for their implementation from internal and external funding sources (a total of USD 11,000). Conversely, the CHPP plans were implemented and coordinated by seven (7) community trainees, with a total of 224 individuals impacted in communities who reported statistical significance in the change in knowledge related to the topic presented (*p* < 0.05). One of these educational plans received financial support from a private institution for its implementation (USD 7500/3 years).

A strategy we had not implemented in previous studies was the creation of a community forum as a space designed for two-way communication between trainees and academic researchers from various fields of health, such as psychology, public health, and medicine. This forum allowed trainees to present their research questions and objectives, which reflected their communities’ health concerns. This approach aligns with ensuring that research topics come from the community, as opposed to traditional research frameworks, in which the researcher presents predetermined topics or already designed projects to the communities [46]. This dynamic promotes a more equitable collaboration focused on the real needs of the population, fostering a constructive dialogue in the search for mutual benefits and encouraging healthy community–academic partnerships.

An additional noteworthy aspect to highlight is that some of the pilot projects proposed by the trainees anticipate integrating health education to reduce risk factors associated with problems identified in the community. This aligns with our original purpose: to create a novel formal academic platform for the Hispanic population (CTIHD) that allows the combination of research and education in health, aiming at developing models that offer an integrated and multilevel response to community health problems. Other programs have been implemented with valuable results; however, most of them typically focus on either research or community health promotion separately [32,39,40,41,46,47,48,49,50,51,52,53].

Lastly, some studies have expressed concerns or controversies about initiatives that seek to train community in research [32]. Proponents of this view argue that this type of intervention fosters the transformation of community members into individuals as academic researchers, which is seen as a loss of identity as community representatives. Our experience during the development process and the early curriculum implementation results differs from this view. The two-way communication we have maintained with our community partners and the continuous feedback from community trainees and alumni confirm that this type of training, rather than distancing them from their role as community leaders, empowers them to continue to serve as transformative social entities. For example, during the final briefing, a trainee from the community expressed, “… we do not stop being leaders. On the contrary, studying allows us to understand more about what is happening in our communities. Now, we have better tools to continue fighting for our people. In addition, through research, we can serve communities”.

The impact and purpose of the CTIHD are perhaps best illustrated by the words of the trainees themselves. Their testimonials highlight the program’s impact in empowering community and bridging the gap between academic research, health promotion, and community needs:
“I think this program opened a lot of doors for us, it let us know that we can reach out to the community and work in collaboration and partnerships. In addition, we can offer our services with credibility. This is very broad and prepares us to feel safe. When we go to present our information, we feel safe. This program has empowered us.”(CRP trainee)
“This is the ’medicine’ that the country needs. Because there is much misinformation, we must take the first step and educate.”(CHPP trainee)

These testimonials underscore the transformative potential of CTIHD, showing how it equips the community with the tools and confidence to address health disparities effectively while maintaining their crucial roles as community leaders.

This study has several strengths. It was developed within a health-focused academic institution, integrating researchers and community members trained in CBPR. This integration facilitated the use of a common language for the co-development of pilot study proposals aimed at various priority health issues for the community. It also helped identify resources to implement research proposals and community health promotion plans. In addition, the interdisciplinary and collaborative approach of CTIHD, aligned with CBPR principles, facilitated the integration of experienced researchers from various fields (psychology, public health, medicine, basic sciences, and education). Upon completing all the requirements, 42 trainees received a certification of participation from an academic institution (i.e., PHSU). Using mixed methods to evaluate the CTIHD also provided a more comprehensive and synergistic understanding of the program outcomes and community trainees’ experiences. Participants accepted into the program represent various groups and sectors, providing multiple perspectives and knowledge of class dynamics and developing research proposals and community health promotional plans.

On the other hand, the curricular content of the programs seeks to promote and stimulate critical thinking by integrating not only the principles of CBPR but also other courses such as Social Determinants of Health, Ethics in CBPR, and Translational Research. A community researcher and health promoter must be able to recognize that communities’ health problems can be associated with their reality, resources, and living conditions. Some topics identified by the community trainees to develop their pilot proposals and educational plans reflect this vision (e.g., access to health services, bullying, social violence).

This study is subject to some limitations. The results are not generalizable as participants were recruited from the southern region of Puerto Rico using non-probabilistic (purposeful) sampling, emphasizing the need to consider the study’s specific context. Moreover, pandemic-related challenges, such as difficulties in engaging students and maintaining active participation during online classes, along with pandemic-induced stress, might have influenced the focus and academic performance of community trainees while influencing the practical experiences of the programs [54]. Additionally, some participants could have been excluded from applying and participating due to a lack of technology or internet access. To remediate this, CTIHD provided technological support to students by lending laptops as needed. To improve the retention of participants in the future, refinement of inclusion criteria (e.g., time availability) and recruitment processes will be evaluated. Additionally, difficulties were observed in engaging and recruiting more men in the programs.

## 5. Conclusions

The CTIHD was successfully implemented using a curricular design framed by the competency-based learning model. The structure, sequence, and content were refined through an iterative quantitative and qualitative evaluation process. The results of the CTIHD’s implementation over the past five years include: forty-two (42) community members completed the established requirements and received certification, seven (7) alliances were established between trainees and academic researchers, seven (7) research proposals were co-developed, and twenty-two (22) health promotion plans were designed. These outcomes support that this community training model framed in the principles of CBPR can be effective in promoting the development of community–academic partnerships and promoting health disparity research. Additionally, results exceeded expectation with three (3) proposals and one (1) educational plan being implemented through internal and external funding mechanisms. Formal training of the community expands opportunities for their active participation in all stages of the research process and facilitates the development of community-led studies. The high retention rate of trainees (83%) suggests that this training opportunity was well-accepted by the community. Creating formal platforms to promote community capacity building represents a way to empower community members, encourage critical thinking, prepare them for social action, and advance CEnR.

## Figures and Tables

**Figure 1 ijerph-22-00080-f001:**
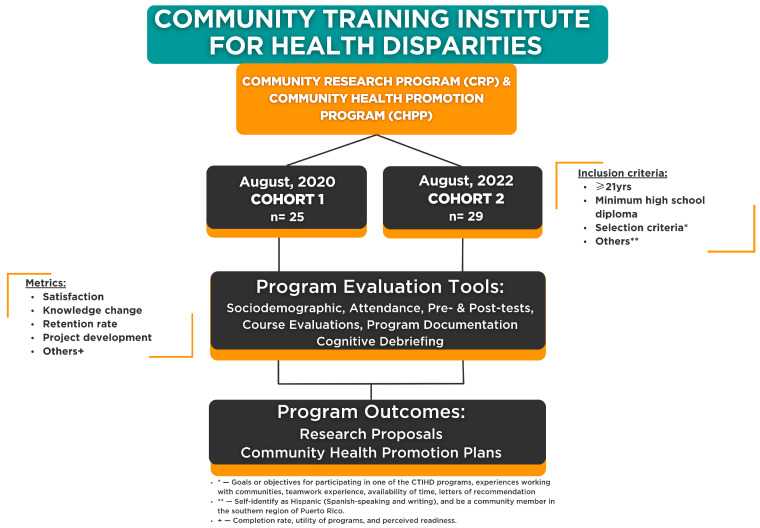
Overall CTIHD program structure.

**Figure 2 ijerph-22-00080-f002:**
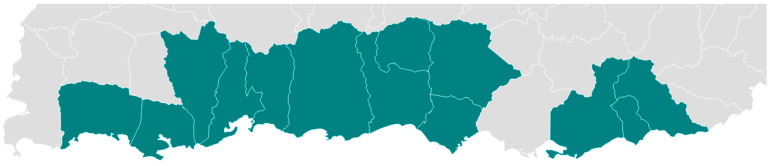
Catchment area by municipality of CTIHD trainees recruited. Note: municipalities of Southern Puerto Rico included in the catchment area of the CTIHD where trainees were recruited are Arroyo, Coamo, Guánica, Guayama, Guayanilla, Juana Díaz, Lajas, Patillas, Peñuelas, Ponce, Santa Isabel, Villalba, and Yauco.

**Table 1 ijerph-22-00080-t001:** Community trainees’ sociodemographic characteristics from two cohorts (n = 43).

Variable		n ^a^	%
Sex			
	Male	5	12
	Female	38	88
	Total	43	100
Age, Years			
	24–34	7	16
	35–44	7	16
	45–54	10	23
	55–64	16	37
	65 or more	3	7
	Total	43	100
Marital Status			
	Single/never married	11	26
	Married	17	40
	Cohabiting with partner	8	19
	Widowed	6	14
	Divorced/separated	1	2
	Total	43	100
Annual Family Income			
	USD 9999 or less	6	14
	USD 10,00–$14,999	8	19
	USD 15,000–$24,999	2	5
	USD 25,000–$34,999	7	16
	USD 35,000–$49,999	8	19
	USD 50,000–$74,999	6	14
	USD 75,000 or more	6	14
	Total	43	100
Education			
	Complete HS	1	2
	Technical degree	1	2
	Associate degree	8	19
	Bachelor’s degree	16	37
	Master’s degree	14	33
	Doctorate degree	3	7
	Total	43	100
Hispanic Origin			
	Yes	43	100
	Total	43	100
Race			
	White	21	49
	Black or African American	15	35
	More than one race	7	16
	Total	43	100

Note. ^a^ Data from eleven participants are not available. n = number of participants; % = percentage.

**Table 2 ijerph-22-00080-t002:** Change in knowledge: pre- and post-test percentage results for cohorts 1 and 2 and pooled of the CTIHD.

Program	Cohort	Group	n	Md	r	Z	Sig.
Community Research	1	Pre	63	50	0.28	3.24	0.001
		Post	63	60			
	2	Pre	70	55	0.40	4.76	<0.001
		Post	70	65			
	Pooled	Pre	133	55	0.20	3.24	0.001
		Post	133	60			
Community Health Promotion	1	Pre	166	60	0.34	6.26	<0.001
		Post	166	71			
	2	Pre	291	67	0.43	10.51	<0.001
		Post	291	83			
	Pooled	Pre	457	64	0.41	12.26	<0.001
		Post	457	79			

Note. N = observations in data set; MP = Median Percentage; r = Effect Size (Cohen’s d); Z = z-value; Sig. = asymptotic significance (2-tailed).

**Table 3 ijerph-22-00080-t003:** Overall CTIHD course evaluation criteria by cohort and program.

Criteria (Summary of Questionnaire Measurement Items)	Cohort 1		Cohort 2	
	CRP (%)	CHPP (%)	CRP (%)	CHPP (%)
Good Course Structure and Organization (clear learning objectives, adequate difficulty level of content, relevant topics, logically organized, precise requirements)	98.5	93.8	97.2	100
Appropriate Course Learning Experiences (appropriate activities and learning experience, adequate time provided, practical applications, enjoyable learning experience, promotion of independent learning, instructor-facilitated learning experience, helpful supplementary resources)	100	93.8	98.6	96.7
Instructor Evaluation (clear communication, the appropriate level of complexity for explanations, visual aids to facilitate teaching, active participation in class, relevant materials and references, respect, and consideration, questions were addressed, positive learning environment)	98.4	93.8	100	100
Course Evaluation and Instructor Feedback Criteria (Scoring criteria, fulfilling activities requirements, student progress, timely and constructive feedback, multiple assessment indicators).	98.3	88.2	97.2	100
Satisfaction with Virtual Tools Used (ease of use of virtual meeting platform, easy access to course documentation, usefulness of virtual communication channels [e.g., email, chat groups])	98.3	93.8	100	100
Course General Satisfaction (adequate group interactions, ease of understanding topics, course expectations were met, increased my desire to know more about research or health promotion, overall satisfaction)	100	93.8	100	96.7

Note. Percentages (%) include agree and totally agree. CRP = Community Research Program; CHPP = Community Health Promotion Program.

**Table 4 ijerph-22-00080-t004:** CITHD end of program cognitive debriefing—Community Research and Community Health Promotion Program.

Final Debriefing Analysis of CTIHD	
Evaluated Criteria/Questions	
ACCEPTANCE: General recommendations for the program (program structure, order in which the courses were taught, balance between didactic activity and practical activity, schedules, professors, materials offered)	
Themes/Verbatim	
Community Research Program	Community Health Promotion Program
Adequate Course Content A+, extraordinary courses. We had vast and, at the same time, concrete training, which was adequate. Improve Course Content The research course must be given a good update, looked at more closely, expanded, and focused more on qualitative and triangular research, such as using focus groups and interviews, among others. Course Length As for the research methods professor’s class, it is an excellent resource, but the course she offered needs more time. Practical Activities … The teachers encouraged participation, and the teacher’s review before class was perfect…. Research methods professor gave us more reading than practice, so the time was very limited for us. Research Proposal Time We must give more time to develop the research topic related to the communities where we work. Course Schedules The schedule may be set to two more days but fewer hours. The schedule for me was good. The schedule could be from 6:00 pm for those who work. Virtual vs. Presential It seems to me that the hybrid modality would be ideal; this allows people who are not from Ponce to participate in this training. … Taking this into account, all the courses were through ZOOM, and many times, we worked as a team, and it became difficult to develop them.	Adequate Course Content I liked the variety of topics and instructors in Course 1. In general, the topics and content of the program were very beneficial. I thought the themes were perfect and the resources excellent. Amid everything, those who knew gave us what they could in the best way … in terms of the staff, the chosen topics, and the material offered, I found it to be perfect and complete. For me, the course puts things in order, which is very beneficial for those who want to make action plans for the community. Technical Language Some topics are very abstract, and one must understand them gradually. And they must explain them better because doubts remain. Practical Activities Incorporate more group activities and practices during the sessions and incorporate the dynamics of implementation (role-playing) during the sessions. Incorporate the creation and development of educational material during the sessions. I would have preferred more practical activities in both courses. Adequate Course Schedules The schedule was comfortable and practical for me. Saturdays were comfortable for me. Virtual vs. Presential I would have liked the sessions to be in person. Otherwise, I feel satisfied. The remote format was very beneficial to me since I live outside of Ponce, and it’s difficult for me to travel. I would have liked the hybrid format as well as in person. It would be great if both options were offered.
PROJECT DEVELOPMENT: What do you think about preparing the research proposal or educational plan?	
Community Research Program	Community Health Promotion Program
Improve Mentorship I think that in the future, students should work closely with a mentor who is available for that first chapter (literature review) and can always help them. Academic Stress A lot of headaches and a lot of stress; we didn’t have a review before starting, although our researcher helped us. I think that many things were asked of us for which we were not prepared; it was as if it were a doctoral-level job. Adequate Community–Academic Partnership My experience of working with academic researchers was excellent. My experience with your academic researcher was also good. Excellent, but I have to say that my team was excellent. Recommendations They must do a review before starting the research proposal. But it does seem that in the proposal, we had some resources (although the researcher was wonderful, she had limits due to time constraints), almost like it was like a class to meet. Because we fell into disuse … I think that from the first course, we should be guided in what our research topic could be and that when we reach the last ones, we will not be desperate for time.	Adequate Mentorship I liked the feedback process of drafting the educational plan. Improve Mentorship It was difficult for me to prepare the educational plan in terms of structure to determine how to write it. I would have liked more direction. Adequate Project Development Process I felt that the implementation process was enriching as a practical activity. Improve Project Development Process It was not previously discussed how an educational plan is developed during the sessions but at the end of the courses. The discussion element would have been beneficial for understanding the elaboration. Recommendations My recommendation for the design of the educational plan: I think we need a “hands-on” session where we are there, like in a laboratory, where we can maybe give each other support creating those plans right there live. Well, we take all the information with us, and when it’s time to sit down, it gets complicated. Possibly, by giving each other group support, we can get a better version of what we had planned.
PERCEIVED READINESS: How prepared do you feel to collaborate in a research project as a community researcher or an educational plan as a community health promoter?	
Community Research Program	Community Health Promotion Program
Adequate Readiness I reaffirm that we are ready and prepared. Simple, I think we already have the tools; we know how to use them. Now, we must use them. In other words, I believe that actual preparation begins now and that we should not be afraid; that is, I think that experience is acquired by doing. We have a lot of knowledge we did not have when we started, but we must continue strengthening ourselves in many areas that could have left us wanting more enrichment.	Adequate Readiness I feel very prepared. I’m already doing it. My proposal is for an organization with a community program where I am a leader. I consider that with the preparation we have obtained, we each have our skills. It reflects how safe you have made us. I believe that the course encourages this. I believe that this experience that we have acquired, in one way or another, has taught us to be more confident and secure and to know that the team is at the back supporting all of us. For me I understand that the program has prepared me a lot. I have spent several years coordinating activities in some communities. This program has helped me strengthen my knowledge and be able to work with communities. The program’s courses gave me the strength I needed to keep going. I have many plans; I want to do many things in my community and town. We know that through the action plan, I can receive the help I need.
PERCEIVED READINESS: How prepared do you feel to guide a person in the community who wants to carry out some research in the community or an educational activity to promote health?	
Community Research Program	Community Health Promotion Program
Adequate Readiness I believe we have the tools. I think we can bring the information because we are all community leaders, and we want our communities to be involved. I am a faithful believer, and I always say that research is for the community, that the community must know what is happening in the Academy, that the Academy must be within the communities, and that it is a joint effort. I believe that you need us; we need you. As a person, I feel prepared because I work in communities and have had the opportunity to have good mentors. As I mentioned recently, I bet this cohort is prepared because we have all enriched ourselves in one way or another. I hope that in the same way that it is true for us, it is also true for academic researchers, and they are prepared to join the community because this is a team; this is teamwork.	Adequate Readiness I feel prepared to educate a community and bring education to rehabilitation. With the training received, I feel prepared to educate since we could do so by offering training. The program gave me a preamble to touch on other issues in the community, where I directed a project and developed an educational plan to educate different organizations and identities that offer services to the older adult population.
PERCEIVED UTILITY: What is your opinion about the contributions that this program can give to community situations or problems?	
Community Research Program	Community Health Promotion Program
Adequate Utility This is the medicine that the country needs because there is a lot of misinformation, so we must take the first step. If we are pioneers, we must not overlook this group and let us not miss the time when this was presented in the newspapers. The community leader is the social structure this country has left to get things done in many areas. What a leader does not do, will not be done—“Novel and pioneering program with community leaders”. … The things of the community belong to the community, and the community must solve them. Ultimately, this also becomes a pretext for the community to understand that it is empowered and does not have to wait for things to arrive. It would sound ugly to me to say it, but we can’t wait for things to fall from heaven. … Through this program, I could not realize all the mistakes in the education department; we can point them out, identify them, and be willing to address them and make a difference with them. You have educated us in such a way that we identified when those public policies regarding bullying came out from the education department. We made direct points and identified them, so I am infinitely grateful for this. Thanks to this proposal, we are going to do it; we are going to the Department of Education to contribute significantly to reducing bullying. I am infinitely grateful, thank you. Live it, be passionate about it. I always see, listen, and criticize myself because we see a problem, but we leave it there. We, as professionals, need to act against that problem and help the community move forward. I believe they have already given us tools, and it is time to begin taking control and living it. The communities should become passionate to get out of the problem or situation. It is essential to tell the community that they must support these projects because this will benefit the community. We must make alliances with the communities. Marketing the Program The program needs to be marketed.	Adequate Utility They are opening the door to very current and relevant topics. This is a good step to bring information closer to the communities and the leadership representing them. As this program has done, we must integrate more community leaders into this process. This program can be the beginning of many projects; the health topic covers a lot. More people are needed to start this process. This program serves as an excellent basis for many projects. I believe this program opened many doors for us, letting us know that we can reach the community, collaborate, and make alliances. And that we can offer our services with credibility. This is very broad and prepares us to feel safe. When we go to present our information, we feel secure. This program has empowered us. I understand that if it provides super-essential information, we often have many ideas and don’t know how to put them on paper to make the project sustainable so that we can explain them to people so that they can understand them.
OVERALL SATISFACTION: In general, how satisfied are you with the activities offered by the program?	
Community Research Program	Community Health Promotion Program
Adequate Satisfaction I feel much more prepared, and I learned a lot. I feel very proud of myself. Great satisfaction with a unique group and thankful for all the knowledge shared between colleagues. I am very satisfied, and I am going to make an analogy because I felt like when one is climbing a mountain. It is a mountain that one wants to climb. I was enjoying things, but I also felt tired. But I was looking up and overcoming the tiredness, and when I felt that I had arrived, it gave me a feeling that the time invested, the difficulties that we had, the achievements, they were all worth it, they are worth it, and they would be worth it. Very grateful for the program. I am very satisfied, and I think that part of any curriculum or program is what they do today. Because there are always things to improve, they are collecting ideas to improve the following curricula, which I think is excellent. I am also satisfied; I have learned, and I have tools that I did not have before. Very satisfied that they are open to recommendations. which is of great value for forming a program.	Adequate Satisfaction I feel delighted and grateful for the opportunity. I feel grateful and happy. I am very happy and satisfied. The program met my expectations. I feel very happy for the support and grateful for the process and the challenges overcome. In general, it has been diverse; we have had different themes. I believe that we have been able to create a community among ourselves.

**Table 5 ijerph-22-00080-t005:** Community Training Institute for Health Disparities’ program outcomes.

Community Research Program	Community Health Promoters Program
Seven (7) community–academic partnerships formed, and research proposals developed and: Community training in assertive communication among Latinx ^a^ coping with cancer (TALC). Overweight and obesity in men and women aged 21 to 70 years from disadvantaged communities in Ponce and Villalba. Levels of anxiety and depression in residents of a community in Ponce, Puerto Rico. Exploring needs related to school bullying management among adolescents. The impact of lack of access to health services in disadvantaged areas and their social determinants in participants with type 2 diabetes. Socio-emotional profile of adolescents in a community. Clinical anxiety and depression and mechanisms of coping in a group of men over 50 years old who have received a cancer diagnosis from Ponce and Villalba Municipalities.	Twenty-two (22) community health promotional plans were developed (a selection): Childhood Diabetes: Building your nutritional plate. Anxiety and Cigarette Smoking. Let’s talk about breast cancer. Access and needs for medical plans. How can you prevent diabetes? A look at violence between young people. Sexual health in young adults We are what we eat! Disaster planning and recovery. Mental Health: Social Isolation and Loneliness.

Note. ^a^ A person of Latin American origin or descent (used as a gender-neutral or nonbinary alternative to Latino or Latina).

## Data Availability

The data presented in this study is available on request from the corresponding author.

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
