# Peer review of "Community Training Institute for Health Disparities: Outcomes of a Formal Opportunity for Community Capacity Building to Increase Health Equity in Southern Puerto Rico"

_ijerph, 2025, doi:10.3390/ijerph22010080_

Round 1
Reviewer 1 Report (New Reviewer)
Comments and Suggestions for Authors
The manuscript CTIHD: Outcomes of a formal opportunity for community capacity building to increase health equity is well thought out and well written.
I have only minor comments, yet some will be important for communicating with this publication's global readership.
1. I would add one more keyword, either Puerto Rico or Hispanic, because the study is limited to Puerto Rico.
2. The title could also be “… to increase health equity in southern Puerto Rico.”
3. The Introduction is very nicely done. Indeed communities need to be capacitated to participate equitably. This topic is very important and you are presenting a well thought solution and evaluation.
4. In line 73, the sentence “However, other factors to consider are poverty …” The word “however” is unnecessary because you add the idea of other factors not part of an individual’s health to the previous sentence.
5. Line 90, US is not necessarily understood outside of the US, so state United Sates of America.
6. In the Introduction, before Background, add a sentence explaining that the CTIHD’s catchment area is in southern Puerto Rico. What is the language spoken? Spanish or English? Were all participants fluent English speakers? Your audience is global, and they need to know this.
7. Line 214 refers to the verification of the comprehensibility of language. Can you explain this? Is it because they were Spanish speakers, and the work happened in English?
8. Line 254. … and best strategies to [?] data. Is this a typo?
9. Why did the community trainees work individually for the Community Health Promotion Programme? Is it because they conduct their work individually? How was the support from the public health practitioners provided? What did the learners share with the public health practitioners? You describe more for the Community Research Program.
10. In Table 1. 12% are males and 88% are females. Does this reflect the percentages of males and females in the workforce? You refer to this in the last sentence of the discussion where you state that it would not affect the analysis of the results. However, there is no indication that you conducted a gender-based analysis of the data, so it is impossible to state that last sentence as it is in the discussion.
11. Figure 2, description. I count 13 names of municipalities, but the map shows 12. Is this right? My screen could be better, and I could be wrong.
12. Lines 434 to 450. Nicely brought together. However, line 436 starts by saying “Future directions include …[actions] “ and I find these directions are “possible or suggested future directions.” The suggested or possible actions would then stated as “CTIHD will aim to [action]” instead of “CTIHD will [action].”
13. The term Latinx needs to be explained to audiences outside of the United States of America.
Overall, it is a needed and well-written manuscript. Congrats!
Author Response
Reviewer 1:
Open Review
( ) I would not like to sign my review report
(x) I would like to sign my review report
Quality of English Language
(x) The quality of English does not limit my understanding of the research.
( ) The English could be improved to more clearly express the research.
|
Yes |
Can be improved |
Must be improved |
Not applicable |
|
|
Does the introduction provide sufficient background and include all relevant references? |
(x) |
( ) |
( ) |
( ) |
|
Is the research design appropriate? |
(x) |
( ) |
( ) |
( ) |
|
Are the methods adequately described? |
(x) |
( ) |
( ) |
( ) |
|
Are the results clearly presented? |
(x) |
( ) |
( ) |
( ) |
|
Are the conclusions supported by the results? |
(x) |
( ) |
( ) |
( ) |
Comments and Suggestions for Authors
Thank you note to the reviewer: Thank you for your comments and suggestions. We appreciate them, as they help us improve this manuscript and stimulate essential discussions for our work.
The manuscript CTIHD: Outcomes of a formal opportunity for community capacity building to increase health equity is well thought out and well written.
I have only minor comments, yet some will be important for communicating with this publication's global readership.
- I would add one more keyword, either Puerto Rico or Hispanic, because the study is limited to Puerto Rico.
- Line 33 - Hispanic keyword has been integrated
- The title could also be “… to increase health equity in southern Puerto Rico.”
- Line 4 – suggestion has been integrated,
- The Introduction is very nicely done. Indeed communities need to be capacitated to participate equitably. This topic is very important and you are presenting a well thought solution and evaluation.
- In line 73, the sentence “However, other factors to consider are poverty …” The word “however” is unnecessary because you add the idea of other factors not part of an individual’s health to the previous sentence.
- Line 74 – “however” has been removed.
- Line 90, US is not necessarily understood outside of the US, so state United Sates of America.
- Line 91 - Abbreviation has been spelled out.
- In the Introduction, before Background, add a sentence explaining that the CTIHD’s catchment area is in southern Puerto Rico. What is the language spoken? Spanish or English? Were all participants fluent English speakers? Your audience is global, and they need to know this.
- Line 129 – A sentence has been integrated to inform the audience of the catchment area and the language spoken, predominantly Spanish. Not all participants are fluent in English.
- Line 214 refers to the verification of the comprehensibility of language. Can you explain this? Is it because they were Spanish speakers, and the work happened in English?
- Line 217 – Comprehensibility, in this sentence, refers to how questions were written to ensure they are not vague or ambiguous. The word “language” was removed to avoid confusion, and the following was added: “i.e., ensure no vague or ambiguous questions.”
- Line 254. … and best strategies to [?] data. Is this a typo?
- Line 259 – Yes, it’s a typo. It has been corrected to “collect”.
- Why did the community trainees work individually for the Community Health Promotion Programme? Is it because they conduct their work individually? How was the support from the public health practitioners provided? What did the learners share with the public health practitioners? You describe more for the Community Research Program.
- Lines 266 to 279. We have updated the description of the program to answer the questions above. Here is the paragraph with the answers integrated: “For the Community Health Promotion Program, community trainees worked individually on developing their community health promotion action plans. This individualized approach allowed each trainee to tailor their plans to address the unique needs of their respective communities. The work was guided by public health practitioners from the RCMI-CEC with extensive experience in health education, who provided critical support, including expert instruction, mentorship, and regular feedback to ensure the plans were practical and impactful. The structure also aligned with the trainees' roles as community health workers (CHWs), where they are expected to conduct their work autonomously and effectively address localized health issues. The individualized format ensured trainees could apply their problem-solving skills in a real-world context while receiving tailored academic support as needed. Trainees shared valuable feedback and insights with the practitioners, contributing to the iterative improvement of the program and fostering personal and professional growth.”
- In Table 1. 12% are males and 88% are females. Does this reflect the percentages of males and females in the workforce? You refer to this in the last sentence of the discussion where you state that it would not affect the analysis of the results. However, there is no indication that you conducted a gender-based analysis of the data, so it is impossible to state that last sentence as it is in the discussion.
- This does not reflect the percentages of the current workforce in the United States of America or Puerto Rico. We will delete that last statement related to the gender-based analysis.
- Figure 2, description. I counted 13 names of municipalities, but the map shows 12. Is this right? My screen could be better, and I could be wrong.
- We have updated the figure to include the missing municipality (Arroyo).
- Lines 434 to 450. Nicely brought together. However, line 436 starts by saying “Future directions include …[actions] “ and I find these directions are “possible or suggested future directions.” The suggested or possible actions would then stated as “CTIHD will aim to [action]” instead of “CTIHD will [action].”
- Line 450 – We have updated the start of this sentence with “CTIHD will aim to [action]” .
- The term Latinx needs to be explained to audiences outside of the United States of America.
- Table 5 - A definition has been integrated as a note in Table 5 for this term.
Overall, it is a needed and well-written manuscript. Congrats!
Reviewer 2 Report (New Reviewer)
Comments and Suggestions for Authors
The abstract starts from the premise that ‘integration of the community into health research through community-engaged research [CEnR] has proven to be an essential strategy for reducing health inequities.’ The authors declare that ‘it brings significant benefits by addressing community health concerns and promoting active community participation in research.’ They also argue that ’the Community Training Institute for Health Disparities (CTIHD) was established to support this integration based on Community-Based Participatory Research (CBPR) principles.’
Likewise, they clarify that ‘the CTIHD trained two cohorts of community members (n=54) in health disparities research and health promotion to foster Community-Academic Partnerships (CAP) and develop community-led health promotion interventions.’ However, the main objective of the research is only clearly specified on page 3, which is, ‘to report the CTIHD program's implementation, evaluation, and outcomes from the first two cohorts.’ Its guiding question is also not sufficiently aligned with the study at hand, as the article abstract states, ‘the CTIHD curriculum was designed to be problem-centered and competency-based.’
Anyway, the topic is relevant to the scope of the journal, with some interest for readers. Nevertheless, it should be noted that the originality is relatively low given the similar and/or complementary content published by most authors over the past two years. Therefore, there remains doubt whether there are still significant knowledge gaps being filled and about the real significant additions of the article to the subject area compared with other published material. On the other hand, all the 43 remaining participants in the study (nine dropouts were evidenced during the implementation phase, resulting in an 83% overall retention rate) are of Hispanic origin, which also reduces the breadth of the results.
The initial part of the first section (Introduction) contextualizes general aspects, notably of CenR and CBPR, while the second section presents theoretical frameworks based on principles such as: ‘1. Making health equity an issue that represents a shared social value. 2. Increasing community capacity and participation in promoting positive health outcomes. 3. Increasing multisectoral/multi-level collaborations as critical strategies to reduce health inequalities and improve community health outcomes.’ These approaches are related to almost 75% of the references, with recent publication (around 62% dated from 2020 to 2024 and, with these, approximately 94% from the last decade).
The third subsection of the introduction refers to the background of the CTIHD, starting with its goals and design. It also includes ethical approval protocols for the study, which are reinforced in the section on funding.
Preliminarily, the second section (Methods: Program Implementation And Evaluation) presents the conceptual model and program design of CTIHD programs (capacity building coursework, community trainees recruitment and sampling, evaluation and data analysis: knowledge change, satisfaction, retention rate, completion rate, and project proposals, research proposal or community health promotional plan development).
In general terms, the methodological steps are adequately detailed for reproducibility. However, it is essential to highlight the need for differentiating procedures from those adopted in previous studies published by the same team of authors.
In its first part, the third section (Results) shows sociodemographic data (sex, age, marital status, annual famile income, education, hispanic origin, and race) from the two cohorts, sendo os 43 trainees pertinentes a 13 municipalities. V with the 43 trainees coming from 13 municipalities. It is worth noting that previous studies by the same authors mention a smaller number of these administrative divisions.
In the second part of the same section, the data demonstrate that ‘a statistically significant increase in knowledge gained by participants in both individual cohorts and pooled results.’ In the following, ‘satisfaction [course evaluations] was collected quantitatively and qualitatively. In the fourth subsection of results, there is a detailed final debriefing analysis of CTIHD. It is also presented that the retention and completion rates were 83% and 78%, respectively. Both cohorts demonstrated a significant increase in knowledge (p<0.05), and overall satisfaction exceeded 90%.
Additionally, it is specified expected outcomes and implementation of CTIHD pre-pilot proposals and community health promotional plans. According to the authors, ‘outcomes include seven community-academic partnerships, leading to the co-development of research proposals, three of which received funding. Additionally, twenty-two community health promotion plans were developed, with seven implemented, impacting 224 individuals.’ It is essential to highlight the need for differentiating the results from those found in previous studies published by the same team of authors.
The fourth section (Discussion) repeats some of the previous results, presenting several personal statements. However, the scientific debates with the theoretical bases are relatively limited. In this context, the references correspond to only 25% of the total and lose part of their contemporaneity (less than 80% from the last 10 years). It is essential to highlight once again the need for interpreting the results of the article in comparison with those found in previous studies published by the same team of authors.
Findings from the study suggest that ‘the CTIHD effectively provided capacity building, promoted the formation of CAP, and increased community-led health promotion interventions, thereby advancing health disparity research and community initiatives.’ These considerations are reinforced in the fifth section (Conclusions), but in an excessively concise manner. They are consistent with the evidence and arguments presented, but the lack of a main question posed generates some uncertainties about the depth of scientific scope of the manuscript.
At first glance, the references are appropriate. Just over 89% are from the last 10 years (almost 61% from 2020 to 2024). It is worth noting that the cited texts by the same team of authors are mentioned as follows: 'other articles discuss these programs' conceptualization, development, and implementation [...].' It is also interpreted that tables and figures are proper.
Finally, it is suggested to minimize the excessive repetitiveness of the same terms in sentences and paragraphs. These are sometimes very long, making textual comprehension difficult.
Author Response
Reviewer 2:
Open Review
( ) I would not like to sign my review report
(x) I would like to sign my review report
Quality of English Language
(x) The quality of English does not limit my understanding of the research.
( ) The English could be improved to more clearly express the research.
|
Yes |
Can be improved |
Must be improved |
Not applicable |
|
|
Does the introduction provide sufficient background and include all relevant references? |
( ) |
( ) |
(x) |
( ) |
|
Is the research design appropriate? |
( ) |
(x) |
( ) |
( ) |
|
Are the methods adequately described? |
(x) |
( ) |
( ) |
( ) |
|
Are the results clearly presented? |
( ) |
( ) |
(x) |
( ) |
|
Are the conclusions supported by the results? |
( ) |
( ) |
(x) |
( ) |
Comments and Suggestions for Authors
Thank you note to the reviewer: Thank you for your comments and suggestions. We appreciate them, as they help us improve this manuscript and stimulate essential discussions for our work.
The abstract starts from the premise that ‘integration of the community into health research through community-engaged research [CEnR] has proven to be an essential strategy for reducing health inequities.’ The authors declare that ‘it brings significant benefits by addressing community health concerns and promoting active community participation in research.’ They also argue that ’the Community Training Institute for Health Disparities (CTIHD) was established to support this integration based on Community-Based Participatory Research (CBPR) principles.’
Likewise, they clarify that ‘the CTIHD trained two cohorts of community members (n=54) in health disparities research and health promotion to foster Community-Academic Partnerships (CAP) and develop community-led health promotion interventions.’ However, the main objective of the research is only clearly specified on page 3, which is, ‘to report the CTIHD program's implementation, evaluation, and outcomes from the first two cohorts.’ Its guiding question is also not sufficiently aligned with the study at hand, as the article abstract states, ‘the CTIHD curriculum was designed to be problem-centered and competency-based.’
- Delete: We will delete this sentence to reduce confusion and align the study abstract with the main manuscript.
- Line 17-19: The main objective was integrated into the article abstract.
Anyway, the topic is relevant to the scope of the journal, with some interest in readers. Nevertheless, it should be noted that the originality is relatively low given the similar and/or complementary content published by most authors over the past two years. Therefore, there remains doubt whether there are still significant knowledge gaps being filled and about the real significant additions of the article to the subject area compared with other published material. On the other hand, all the 43 remaining participants in the study (nine dropouts were evidenced during the implementation phase, resulting in an 83% overall retention rate) are of Hispanic origin, which also reduces the breadth of the results.
- Line 521: We have considered the limitation of dropouts and will evaluate the inclusion criteria in the future to better retain participants. Additionally, we provided technological support for those with limited access to laptops.
The initial part of the first section (Introduction) contextualizes general aspects, notably of CenR and CBPR, while the second section presents theoretical frameworks based on principles such as: ‘1. Making health equity an issue that represents a shared social value. 2. Increasing community capacity and participation in promoting positive health outcomes. 3. Increasing multisectoral/multi-level collaborations as critical strategies to reduce health inequalities and improve community health outcomes.’ These approaches are related to almost 75% of the references, with recent publication (around 62% dated from 2020 to 2024 and, with these, approximately 94% from the last decade).
The third subsection of the introduction refers to the background of the CTIHD, starting with its goals and design. It also includes ethical approval protocols for the study, which are reinforced in the section on funding.
Preliminarily, the second section (Methods: Program Implementation And Evaluation) presents the conceptual model and program design of CTIHD programs (capacity building coursework, community trainees recruitment and sampling, evaluation and data analysis: knowledge change, satisfaction, retention rate, completion rate, and project proposals, research proposal or community health promotional plan development).
In general terms, the methodological steps are adequately detailed for reproducibility. However, it is essential to highlight the need for differentiating procedures from those adopted in previous studies published by the same team of authors.
- Line 256: A strategy we had not implemented in previous studies was integrating research activities and health promotion initiatives to differentiate from previously published studies. We are establishing the base to create a strategically designed space to combine research with health promotion to respond to community needs, such as the Community Forum.
In its first part, the third section (Results) shows sociodemographic data (sex, age, marital status, annual family income, education, Hispanic origin, and race) from the two cohorts, sendo os 43 trainees pertinent's 13 municipalities. V with the 43 trainees coming from 13 municipalities. It is worth noting that previous studies by the same authors mention a smaller number of these administrative divisions.
- Response: Previous publications focused on one program associated with the CTIHD and did not encompass all municipalities from both programs.
In the second part of the same section, the data demonstrate that ‘a statistically significant increase in knowledge gained by participants in both individual cohorts and pooled results.’ In the following, ‘satisfaction [course evaluations] was collected quantitatively and qualitatively. In the fourth subsection of results, there is a detailed final debriefing analysis of CTIHD. It is also presented that the retention and completion rates were 83% and 78%, respectively. Both cohorts demonstrated a significant increase in knowledge (p<0.05), and overall satisfaction exceeded 90%.
Additionally, it is specified expected outcomes and implementation of CTIHD pre-pilot proposals and community health promotional plans. According to the authors, ‘outcomes include seven community-academic partnerships, leading to the co-development of research proposals, three of which received funding. Additionally, twenty-two community health promotion plans were developed, with seven implemented, impacting 224 individuals.’ It is essential to highlight the need for differentiating the results from those found in previous studies published by the same team of authors.
- Response: The CTIHD comprises two programs: the Community Health Promotion Program (previously presented in an article) and the Community Research Program (not reported in an article). We intend that reporting and disseminating the results of both programs is essential to providing a holistic understanding of the CTIHD's overall impact and outcomes.
The fourth section (Discussion) repeats some of the previous results, presenting several personal statements. However, the scientific debates with the theoretical bases are relatively limited. In this context, the references correspond to only 25% of the total and lose part of their contemporaneity (less than 80% from the last 10 years). It is essential to highlight once again the need for interpreting the results of the article compared to those found in previous studies published by the same team of authors.
- Line 345: The discussion was revised and edited to focus more on the interpretation of results and mentions the differences between previously published materials by the author team.
Findings from the study suggest that ‘the CTIHD effectively provided capacity building, promoted the formation of CAP, and increased community-led health promotion interventions, thereby advancing health disparity research and community initiatives.’ These considerations are reinforced in the fifth section (Conclusions), but in an excessively concise manner. They are consistent with the evidence and arguments presented, but the lack of a main question posed generates some uncertainties about the depth of scientific scope of the manuscript.
- Line 345: The discussion and conclusion sections were revised and edited to be more reflective and critical about how the CTIHD's current impact and outcomes will provide a base for developing a model for attending community health needs to reduce health disparities.
At first glance, the references are appropriate. Just over 89% are from the last 10 years (almost 61% from 2020 to 2024). It is worth noting that the cited texts by the same team of authors are mentioned as follows: 'other articles discuss these programs' conceptualization, development, and implementation [...].' It is also interpreted that tables and figures are proper.
Finally, it is suggested to minimize the excessive repetitiveness of the same terms in sentences and paragraphs. These are sometimes very long, making textual comprehension difficult.
Round 2
Reviewer 2 Report (New Reviewer)
Comments and Suggestions for Authors
The revised article addresses the main observations previously made. As final contributions, it is suggested to provide greater clarity regarding the research question as a whole and to deepen the conclusions, highlighting the evidence and arguments presented and responsive to the main question posed. If possible, it is also recommended to minimize the excessive repetitiveness of the same terms in sentences and paragraphs, by subdividing the latter when they are very long.
Author Response
The revised article addresses the main observations previously made. As final contributions, it is suggested to provide greater clarity regarding the research question as a whole and to deepen the conclusions, highlighting the evidence and arguments presented and responsive to the main question posed.
- Guiding question - Line 130: Thank you for your feedback. We appreciate your suggestion to provide greater clarity regarding the research question and to deepen the conclusions. In response, we have revised our purpose statement to serve as a guiding question for the study: “With the creation of the CTIHD, we planned to establish an innovative and sustainable structure in PR, co-developed with members of the community and supported by a health-focused university. Our purpose was to examine whether the development and implementation of a formal training institute tailored to community members would be an effective strategy for acquiring knowledge and skills that facilitate the formation of partnerships with academic researchers and disseminating health education to peers.”
- Conclusions – Line 540: The conclusions have been deepened, highlighting the evidence and arguments presented and aligned with the guiding purpose (guiding question). The conclusions are as follows: “The CTIHD was successfully implemented using a curricular design framed by the competency-based learning model. The structure, sequence, and content were refined through an iterative quantitative and qualitative evaluation process. The results of the CTIHD’s implementation over the past five years include: forty-two (42) community members completed the established requirements and received certification, seven (7) alliances were established between trainees and academic researchers, seven (7) research proposals were co-developed, and twenty-two (22) health promotion plans were designed. These outcomes support that this community training model framed in the principles of CBPR can be effective in promoting the development of community-academic partnerships and promoting health disparity research. Additionally, results exceeded expectation with three (3) proposals and one (1) educational plan being implemented through internal and external funding mechanisms. Formal training of the community expands opportunities for their active participation in all stages of the research process and facilitates the development of community-led studies. The high retention rate of trainees (83%) suggests that this training opportunity was well-accepted by the community. Creating formal platforms to promote community capacity building represents a way to empower community members, encourage critical thinking, prepare them for social action, and advancing CEnR.”
If possible, it is also recommended to minimize the excessive repetitiveness of the same terms in sentences and paragraphs, by subdividing the latter when they are very long.
- We have reviewed the manuscript and reduced the same terms in sentences and paragraphs. Please let us know if there are any additional recommendations to ensure the best final version.
This manuscript is a resubmission of an earlier submission. The following is a list of the peer review reports and author responses from that submission.
Round 1
Reviewer 1 Report
Comments and Suggestions for Authors
The manuscript presents an initiative of great value for the strengthening of Community-Engage Research, promoting through its CTIHD the training of community researchers with the aim of promoting health disparities research and community health education. Based on this, the results of a training program are presented. However, the manuscript is not eligible for publication for the following reasons.
1. A specific experience focused on the training of community researchers is presented and evaluated, without assessing the impact of this training on the objective of the CTIHD. It is understood that it is too early to assess the impact, so perhaps it is not the most appropriate time to publish in final terms. That is, it is necessary to expand in time and in the participants trained to determine its impact. In fact, one of the results that it presents is the “complete pase”, but this phase is only based on proposals and training plans. It would be interesting to know which of these proposals have resulted in community research.
2. Methodological deficiencies are detected given that in the results section there are qualitative data that have not been previously explained as these types of data will be collected and analyzed later. Furthermore, these qualitative data have been subsequently discussed by the authors, but as they are presented they are not data that can be considered to have been obtained with a rigorous methodological basis, even though they are qualitative, so that what is expressed in the discussion can be understood as an "impression of the authors".
3. There are some formal issues that should be corrected, thus some citations do not appear in the references (8, 9, 10), others appear with different numbering (6 and 16) and as a whole do not follow the same criteria.
Reviewer 2 Report
Comments and Suggestions for Authors
For the improvement of this manuscript, here are some comments and suggestion :
â‘ Abstract:
-The writing is not standard,which needs to be rewritten. The summary of conclusions needs to be fleshed out with the findings of the study.
â‘¡ Methods:
-Inclusion and exclusion criteria were not precisely quantified by description.
1) 21 years 147 or older, 2) minimum of a high school degree, 3) be a community member in one of the following municipalities of the southern region of Puerto Rico.
The community population composition is extremely complex, and the inclusion and exclusion criteria established for this experiment were too broad.
1. Potential for teamwork
2. Potential for developing research or educational skills
3. Motivation, commitment, and emotional stability to participate in all the sessions and activities of the programs
The inclusion and exclusion criteria for the "Potential for teamwork" category should be described below, along with the methodology for measuring this criterion.
-When selecting team members, the text does not explain in detail how the Potential for teamwork, Potential for developing research or educational skills of the members should be determined.
- Representativeness of the sample
Why were participants from these cities selected as community trainees and are they representative.
How the percentage of participants who qualified was determined.
â‘¢ Results:
-Table 2 :
Whether the demographic sociology of the participants, which included only one man, representing only 6%, would affect the analysis of the experiment should be detailed in the article.
Why is the annual income of "$15,000 - $24,999" zero, is there a problem in the experiment, and is it meaningful to include the data of zero?
â‘£Discussion
The results section should include more descriptions of the data to support the inferences with data whose experimental inferences are more convincing.
⑤Conclusion:
The conclusions are too mixed and the summary of the study needs to be further fleshed out with the experimental results.
